# Experimental Assessment of Perhydro-Dibenzyltoluene Dehydrogenation Reaction Kinetics in a Continuous Flow System for Stable Hydrogen Supply

**DOI:** 10.3390/ma14247613

**Published:** 2021-12-10

**Authors:** Sanghyoun Park, Mujahid Naseem, Sangyong Lee

**Affiliations:** Mechanical Robotics and Energy Department, Dongguk University, Seoul 100-715, Korea; psh6851@naver.com (S.P.); mujahid.naseem019@gmail.com (M.N.)

**Keywords:** LOHC, DBT, hydrogen, dehydrogenation, catalyst, kinetics, reaction rate

## Abstract

The development of alternate clean energy resources is among the most pressing issues in the energy sector in order to preserve the global natural environment. One of the ideal candidates is the utilization of hydrogen as a primary fuel in lieu of fossil fuels. It can be safely stored in liquid organic hydrogen carrier (LOHC) materials and recovered on demand. A uniform supply of hydrogen is essential for power production systems for their smooth operation. This study was conducted to determine the operating conditions of the dehydrogenation of perhydro-dibenzyltoluene (H18-DBT) to ensure that hydrogen supply in a continuous flow reactor remains stable over a wide range of temperatures. The hydrogen flow rate from the dehydrogenation reaction was measured and correlated with the degree of dehydrogenation (DoD) evaluated from the refractive index of reactant liquid samples at various temperatures, WHSV and the initial reactant concentrations. Moreover, a kinetic model is presented holding validity up to a WHSV of 67 h^−1^. The results acquired present a range for an order of reaction from 2.3 to 2.4 with the required activation energy of 171 kJ/mol.

## 1. Introduction

Carbon emissions are the primary cause of the drastic climate changes we are experiencing in the form of global warming. Internationally, strict policies are being implemented in order to reduce carbon emissions via decreasing the use of fossil fuels [1,2,3]. On the other hand, the increase in the global energy consumption is continuously increasing, which presses the urge to develop clean energy alternatives [4,5,6]. Hydrogen has become one of the ideal candidates for an alternate clean energy resource since it became known that it can be directly utilized in energy production systems and positively contribute to the energy storage systems. Hydrogen can be produced by many different methods such as electrolysis, fossil fuel reforming, thermolysis, thermochemical processes including water splitting, photocatalysis, photoelectrochemical electrolysis and fossil fuel reforming (see Table 1). Among these methods, the electrolysis and fossil fuel reforming methods are the most widely used methods [7]. Moreover, hydrogen can be generated from fluctuating renewable energy resources such as solar and wind energies and can be stored [8,9,10]. Conventional hydrogen storage technologies involve liquefaction and high-pressure compression. However, there are safety issues and a lack of infrastructure necessary to distribute the high-pressure hydrogen or liquified hydrogen [11]. The storage of hydrogen in LOHC is known as one of the safe methods of hydrogen transport and especially for long-term hydrogen storage systems with a high volumetric and gravimetric hydrogen storage capacity between 4.3 and 12 wt%. The storage capacities of methanol, formic acid, formaldehyde, aqueous formaldehyde and dibenzyltoluene are known to be 12.5 wt%, 4.3 wt%, 6.7 wt%, 8.4 wt% and 6.2 wt%, respectively [12,13,14,15,16]. The dehydrogenation temperature reported in previous studies is between 25 and 400 °C depending on the LOHC materials used [14,15,16,17,18,19,20,21].

Moreover, by the hydrogenation of LOHC (LOHC+), hydrogen can be stored at ambient conditions in the form of hydrogenated LOHC (LOHC+). Furthermore, LOHC (LOHC+) is non-flammable as well as non-volatile which makes it extremely safe for road transport [22,23]. The total cost of the processing and transportation of LOHC is cheaper than the cost of other storage methods because LOHC does not include a distribution cost. LOHC involves a dehydrogenation and hydrogenation process that can reversibly store and release hydrogen. Table 2 shows a comparison of the characteristics of each hydrogen storage method.

LOHC is used as an energy carrier for the dehydrogenation and hydrogenation processes, which can reversibly store and release hydrogen. The stored hydrogen in the form of LOHC+ can be recovered on demand via the dehydrogenation of the LOHC+ and the carrier material can be recovered in the form of dehydrogenated LOHC (LOHC-) [24]. Representative LOHC materials are summarized in Table 3. Dibenzyltoluene (DBT), N-ethylcarbazole (NEC) and toluene (TOL) are well-known LOHCs. In the LOHC dehydrogenation process, a reactor temperature of 200 °C or higher needs to be maintained [25]. In the case of using toluene as an LOHC, toluene undergoes vaporization during the dehydrogenation process because the operation temperature is higher than the boiling point of toluene which complicates the process and requires an additional process for the separation of LOHC into a gas phase [26]. In the case of using dibenzyltoluene as a hydrogen carrier, the dehydrogenation temperature is lower than the boiling point and the vapor pressure is relatively lower than those of other LOHCs [25]. Additionally, dibenzyltoluene is currently used as a thermal oil which costs approximately USD 4.66/kg. In terms of material cost, toluene has the advantages as it costs USD 0.35/kg [25]. However, toluene is toxic and flammable, and many side reactions such as disproportionation and dealkylation occur during dehydrogenation [27,28,29], whereas dibenzyltoluene has a relatively excellent thermal stability and its generation of impurities is less than 0.01% at 270 °C [30]. Consequently, dibenzyltoluene (H0-DBT) has been investigated as one of the ideal candidates for hydrogen storage [31,32] because of its higher hydrogen content and dehydrogenation temperature [26,33].

H0-DBT accommodates nine molecules of hydrogen in its single molecule as shown in Figure 1. Dehydrogenation is the primary reaction in power generation systems as it is responsible for the steady supply of hydrogen [34]. During dehydrogenation, H18-DBT contacts the catalyst where the reaction occurred and hydrogen is produced in the form of bubbles. The reaction rate of this dehydrogenation depends on the characteristics of the platinum or palladium catalysts [30].

Thus, the kinetic analysis of the catalyst unit is required to optimize the operating conditions and to understand the dehydrogenation reaction. The study of reaction kinetics and the optimization of operating conditions for the dehydrogenation reaction of H18-DBT are of paramount importance [35,36,37,38,39]. Various researchers have experimentally studied the dehydrogenation reaction kinetics of H18-DBT and proposed regressed kinetic models using the Arrhenius approach. The generalized form of the model is given in Equation (1):(1)$$r=-\frac{1}{W_{catalyst}}\frac{dn_{H18-DBT}}{dt}=KC_{H18-DBT}^{n}=k_{0}e^{\left(\frac{-E_{a}}{RT}\right)}C^{n}$$
where *W_catalyst_* is the weight of the catalyst, *K* is the rate constant, *C_H_*_18*-DBT*_ is the concentration of H18-DBT, *E_a_* is the activation energy, and n is the overall reaction order.

Bulgarin et al. [35] proposed a kinetic model for the dehydrogenation of moisture containing perhydro-dibenzyltoluene over a temperature range of 287–297 °C using the 0.3 wt% platinum catalyst. Modisha et al. [36] investigated the dehydrogenation reaction rate of perhydro-dibenzyltoluene with various catalysts over a temperature range of 290–320 °C, and proposed the first-order rate expression. However, their previous study used a dehydrogenation reaction temperature of up to 290 °C in order to prevent the thermal cracking of the carrier material (H18-DBT) [36]. Preuster et al. [37] investigated the dehydrogenation kinetics of different LOHC materials in his doctoral studies and proposed a kinetic model for the dehydrogenation of perhydro-dibenzyltoluene with a platinum catalyst over a wider temperature range of 260–320 °C and a pressure range of 1–5 bar using a 100 mL batch reactor. Peters et al. [38] utilized this model in their study of a solid oxide fuel cell powered via hydrogen from the dehydrogenation of perhydro-dibenzyltoluene. Wunsch et al. [39] investigated the dehydrogenation kinetics of perhydro-dibenzyltoluene using a membrane plug flow reactor (PFR) with a Pd–Ag catalyst for obtaining more accurate conversion data at a given concentration. The different models are compiled in Table 4.

At a temperature beyond 290 °C, the thermal degradation of the carrier material renders the low temperature studies more valuable. Moreover, a continuous supply of hydrogen is required at the power generation unit. In this study, the dehydrogenation kinetic model was proposed and it has an applicability over a wide range of temperatures, especially below 290 °C with a high WSHV for a continuous flow reactor that could produce the continued supply of hydrogen from LOHC+. Experiments were carried out using a differential PFR reactor with a 5 wt% Pt/A_2_O_3_ catalyst. Moreover, the essential information required for the design of an industrial grade dehydrogenation reactor is presented.

## 2. Experiment

### 2.1. Materials

#### 2.1.1. Perhydro-Dibenzyltoluene

Perhydro-dibenzyltoluene was prepared via the hydrogenation of dibenzyltoluene, procured from Sasol Inc. (Sandton, South Africa)., by the trade name of Marlotherm-SH. Hydrogenation was performed in a 10 L batch type reactor with an impeller arrangement and a catalyst cage. The operating conditions were controlled at a temperature of 170 °C and a pressure of 70 bar.

The reactor was initially pre-charged with 6 L H0-DBT and 500 g of 2.0 wt% ruthenium–alumina catalyst with a 3 mm diameter. The impeller was driven at a constant speed of 1700 rpms. Ample time of 10 h was given to ensure that maximum hydrogenation was achieved—up to 99.9% hydrogenation—which was verified via gas chromatography and refractive index measurements. Aglient (Santa Clara, CA, USA), GC-7890A, MS-5975C was used for the gas chromatography analysis and its results are shown in Figure 2.

The refractive index varies linearly with the degree of hydrogenation [36,40]. The refractive index was measured using the Nova-tech (Kingwood, TX, USA) Abbe refractometer VEE GEE model C10 within a measurement range of 1.3000–1.7000 nD, with a resolution of 0.0005 nD and an accuracy of 0.0003 nD. Moreover, the density of H0-DBT and H18-DBT was measured using a KEM (Kyoto, Japan) Density meter WBA-505 with a measuring range of 0.00000~3.00000 g/cm^3^, and an accuracy of ± 0.00005 g/cm^3^ with a repeatability of 0.00010. A few important properties of both H0-DBT and H18-DBT are given in Table 5.

#### 2.1.2. Catalyst

The commercial dehydrogenation catalyst was purchased from Sigma Aldrich (Saint Louis, MO, USA), constituting 5 wt% Pt–Al_2_O_3_ powder catalyst. The specific details of the catalyst used are defined in Table 6.

### 2.2. Experimental Setup

The schematics experimental setup design is presented in Figure 3. Nitrogen was used for purging the system prior to experimentation. A controlled flow rate of H18-DBT was supplied to the reactor from where the hydrogen and dehydrogenated carrier material were produced as product. The flow rate of the produced hydrogen was measured and recorded. The liquid product was also tested for its degree of dehydrogenation (DoD).

Implementation of the schematics designed for the PFR reactor shown in Figure 3 was carried out and two identical PFR reactor apparatuses were set up, as shown in Figure 4. H18-DBT was initially stored in the glass container and was pre-heated to 70 °C via temperature-controlled magnetic stirrer and heater. After the reactor reached the thermal equilibrium, H18-DBT was injected into the heating section of the reactor via HPLC pump (FLOM, Tokyo, Japan, Dual pump KP-22) where the reaction temperature was achieved before contacting the catalyst. The reactor was located within the housing of the temperature-controlled electric furnace maintained at the desired reaction temperature.

#### Reactor Description

A U-shaped reactor was located in the housing of the temperature-controlled electric furnace maintained at the desired reaction temperature. The reactor consisted of a heating section and the catalyst bed, as shown in Figure 5. The heating section was made of a ¼ inch SUS 316 tube, which diverged to ½ inch to connect with the catalyst section where the catalyst was filled to a height of 0.2~1 cm. Quartz wool was located at the top and bottom of the catalyst layer to prevent it from flowing out. The temperature of the reactor was measured using K-type thermocouples located at the center, at the top, at the bottom, and at the walls of the catalyst layer, marked as shown in Figure 5.

### 2.3. Procedure and Analysis

The reactor was initially purged twice with nitrogen at 50 kPa for 1 h prior to commencing the experiments. After the reactor temperature reached 250 °C, LOHC+ was injected at a rate of 0.03 cm^3^/min for 10 h. After the reactor volume was fully filled with LOHC+, the hydrogen production was started. The desired reactor temperature was set and run for 2 h. The sampling of LOHC from the reactor was performed after the catalyst layer temperatures remained stable for at least 1 h with a continuous LOHC+ supply.

Experiments were conducted at various temperatures, WHSVs and the initial concentration of LOHC+. The experimental conditions are listed in Table 7. The reactor temperature was directly controlled via the temperature controller. The value for the weight hour space velocity (WHSV) was calculated using Equation (2). Since the catalyst weight is a constant, only the mass flow rate of LOHC+ (${\dot{m}}_{LOHC+}$) is controlled using the HPLC pump:(2)$$\mathrm{WHSV}=\frac{{\dot{m}}_{LOHC+}}{W_{catalyst}}$$

The required initial concentration was obtained by mixing the measured quantities of perhydro-dibenzyltoluene (H18-DBT) with dibenzyltoluene (H0-DBT), verified by measuring the mixture refractive index. The degree of hydrogenation (DoH) is defined as a ratio of the amount of hydrogen stored in the LOHC to the maximum potential hydrogen storage capacity of the LOHC. It can be calculated by Equation (3) via the hydrogen release rate:(3)$$\mathrm{DoH}=\frac{{\dot{n}}_{H_{2},max}-{\dot{n}}_{H_{2},released}}{{\dot{n}}_{H_{2},max}}\;$$

Since this study was focused on the dehydrogenation rate, the results are presented for the degree of dehydrogenation (DoD) that was defined as a ratio of the hydrogen released to the maximum hydrogen storage capacity of the LOHC. By definition, this is the opposite of DoH and can be expressed as Equation (4):(4)DoD=1−DoH

## 3. Results and Discussion

### 3.1. Validation

The DoH calculated from the hydrogen production rate and the DoH measured from the refractive index of the liquid sample were cross-validated as shown in Figure 6. As shown in Figure 6, the trend line between both methods presents a good correlation emphasizing both methods and providing acceptable results.

### 3.2. Catalyst Temperature Distribution

The temperature difference between the center of the catalyst bed and the reactor wall was observed to be minimal, as shown in Figure 7. At the reactor temperature of 250 °C, the difference between the maximum and minimum temperature in the reactor was less than 0.9 °C. However, above 300 °C, the temperature at the wall (TC 4) was 302.4 °C at 300 °C, 313.5 °C at 310 °C and 324.5 °C at 320 °C. The catalyst bed temperature was lower than that of the reactor wall due to the endothermic nature of the dehydrogenation reaction. The temperature difference between the inlet (TC 1) and outlet (TC 3) of the catalyst layer was less than 1.5 °C. Moreover, the temperature difference between the catalyst layer inlet temperature (TC 1) and center (TC 2) was 0.1 °C, signifying that the temperature distribution in the catalyst layer can be ignored. However, the wall surface temperature difference (TC 4 and TC 5) and the middle of the catalyst layer (TC 2) was thought to be due to thermal conductivity. Therefore, it was considered that the experimental results showed that the temperature distribution of the catalyst layer can be ignored.

### 3.3. Effect of WHSV on DoD

Figure 8 shows the effect of WHSV on the degree of dehydrogenation (DoD) for various initial concentrations of H18-DBT in the input stream at fixed temperatures. The data presented were regressed using the logarithmic trend line. Each graph presents the results at a particular control temperature of the catalyst layer from 270 °C to 320 °C. Moreover, each curve presents a specific H18-DBT initial concentration ranging from 0.3 to 0.9.

Increasing the temperature from 270 °C to 320 °C resulted in an increased DoD from 30% to 54% at a constant initial concentration of 0.9 and WSHV of 0.57. At a higher temperature, the number of reactant molecules with an energy greater than the activation energy increased, resulting in a faster reaction rate. Likewise, the higher the initial concentration is, the higher the reaction rate is, which gives a higher DoD. This was due to an increased number of reactant molecules within the same volume, increasing the probability of successful collisions leading to the reaction. Higher temperatures show an acceptable range of DoD at a given WHSV for hydrogen production. However, to ensure the life of the LOHC, the dehydrogenation temperature would be limited to 290 °C for the commercial process. Moreover, an initial concentration below 0.3 cannot produce an adequate amount of hydrogen as its DoD remains below 10% even at very a low WHSV.

Despite increasing the temperature up to 320 °C, the DoD remained limited to 20% which was not acceptable. In order to maintain a higher DoD, the reaction mixture concentration must be kept as high as possible. It was observed from the experimental results that the least initial concentration of 0.7 should be utilized in order to achieve a satisfactory hydrogen release rate resulting in a DoD greater than 30%. One of possible ways of maintaining a higher molar concentration is by the separation of the LOHC+ from the product mixture and recycling it back to the reactant mixture. As the WHSV is increased, the DoD is decreased. The DoD shows an exponential decay with an increasing WHSV since the residence time within the catalyst layer is decreased. Hence, for an enhanced DoD, WHSV must be maintained near the vicinity of zero as much as possible. Physically, this represents the use of a high catalyst volume over a low flow rate of LOHC+ supplied to the reactor. This increases the interaction of the LOHC molecules with the catalyst leading to a higher dehydrogenation rate and thus resulting in a higher DoD.

### 3.4. Kinetics Model for Dehydrogenation

The kinetic model was developed using the data for WHSV of over 2.0 h^−1^ since rapid hydrogen production occurred which might cause the flooding of the LOHC on the reactor if the WHSV is less than 2.0 h^−1^. The reaction rate (r), reaction order (n) and the reaction rate constant (K) can be calculated using the least squares method from the experimental data. In addition, the pre-exponential constant ($k_{0}$) and activation energy ($E_{a}$) were calculated using the Arrhenius approach given in Equation (1).

Figure 9 shows the plot for $\ln\left(C_{0}\right)$ versus ln(−r), the linearized form of Equation (1) over the temperature range of 250–320 °C. The linear equations from the experimental data at each temperature are calculated and listed in the figure, which have an RMS error of 7% while the lower temperatures exhibit an error of less than 3%. The slope signifies the reaction order (n) and intercept corresponds to the ln(K). The reaction order obtained from the experimental data were from 2.35~2.44.

Figure 10 presents the linearized plot for the rate constant (K) to the 1/RT. The slope signifies the activation energy $\left(E_{a}\right)$ of −171.72 kJ/mol. The kinetic model obtained from the experimental data over a temperature range of 250~320 °C is given as Equation (5):(5)$$r=5.4\times 10^{11}e^{\left(\frac{-171.72}{RT}\right)}C_{H18-DBT}^{2.395\pm 0.045}$$

### 3.5. Application of Dehydrogenation Kinetics for Reactor Design

The composition of H18-DBT in the PFR changes at each point along the flow path and the analysis of the PFR can be performed using a molar balance defined in Equation (6). $X_{H18-DBT}$ is a fractional conversion of H18-DBT where 1 represents the initial molar concentration $\left(C_{0}\right)$ and $X_{out}$ represents the difference in molar concentration at the inlet and at the outlet divided by the initial molar concentration. The relationship between $W_{catalyst}$ and $F_{H18-DBT}$ is the design parameter, which physically represents the amount of catalyst needed to obtain the desired dehydrogenation of H18-DBT in the reactor which can be obtained from the integral equation on the right-hand side of Equation (6). The solution to Equation (6) was obtained by the substitution of r with Equation (5) and solving the definite integral for the required outlet concentration. The calculations were performed for different reactor temperatures’ fully hydrogenated LOHC+. The results are presented in Figure 11 and Figure 12:(6)$$\frac{W_{catalyst}}{F_{H18-DBT}}=\int_{1}^{X_{out}}\frac{dX_{H18-DBT}}{r}$$

Figure 11 and Figure 12 present essential information required for the design of a continuous flow reactor. The flow rate and required DoD determine the amount of catalyst weight needed within the reactor. At higher temperatures, the DoD rapidly increased with the maximum rate and then gradually increased at a declining rate according to the W/F ratio. This signifies the optimal catalyst loading within the reactor to achieve the target DoD.

Even though a higher temperature results in a higher DoD, the dehydrogenation temperature was limited to 290 °C due to the stability of the LOHC material. In the presence of the catalyst, the DBT cannot withstand above 290 °C and disintegrates into lighter molecules due to thermal cracking [43]. Comparing 320 °C and 290 °C in Figure 12, the ratio of the final molar concentration and the initial molar concentration at W/F = 500 is 30% at 320 °C and 70% at 290 °C. This means that reaction temperature of 290 °C needs twice as much catalyst than 320 °C to dehydrogenate the same amount of perhydro-dibenzyltoluene. To increase DoD in a reactor with a limited amount of catalyst, the LOHC can be circulated through the small reactor packed with catalyst several times.

## 4. Conclusions

Hydrogen is one of the most appealing candidates as clean energy resource to replace the fossil fuels. However, the safety of its storage and transport is essential for using hydrogen as a primary fuel. Among LOHC materials dibenzyltoluene is an especially ideal candidate that could be used to accomplish this task.

The dehydrogenation kinetics of various initial concentrations of perhydro-dibenzyltoluene in a plug flow reactor were experimentally studied. Moreover, a kinetic model holding validity over a large range of WHSV and reactor temperatures was presented. To calculate the dehydrogenation rate, the sample’s refractive index was cross-verified with the hydrogen release measured via mass flow meter for a certain time interval.

From the experimental results, it was confirmed that the dehydrogenation rate was higher for the higher initial concentrations and higher temperatures. The kinetic model was proposed to hold validity over temperatures of 250~320 °C and a WHSV greater than 2 h^−1^ and a reaction order ranging from 2.3 to 2.4. In the model, the corresponding activation energy of +171 kJ/mol was obtained. The calculations required for the reactor design were performed by applying the derived equation. Higher temperatures have the advantage of reducing the amount of catalyst required, but the disadvantage is that the lifetime of the perhydro-dibenzyltoluene can be reduced. For the effective operation of the dehydrogenation reactor at an appropriate temperature, physical and economic considerations according to the increase in the amount of catalyst or a process capable of reducing the amount of catalyst should be considered.

## Figures and Tables

**Figure 1 materials-14-07613-f001:**
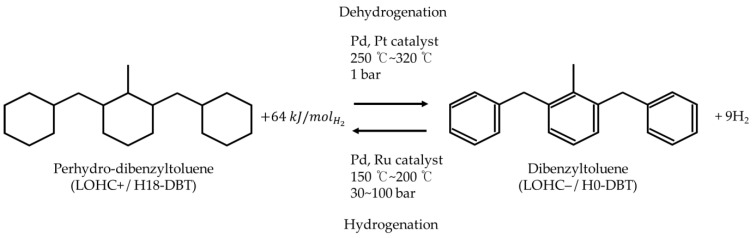
Schematic diagram of dehydrogenation and hydrogenation of dibenzyltoluene/perhydro-dibenzyltoluene [34].

**Figure 2 materials-14-07613-f002:**
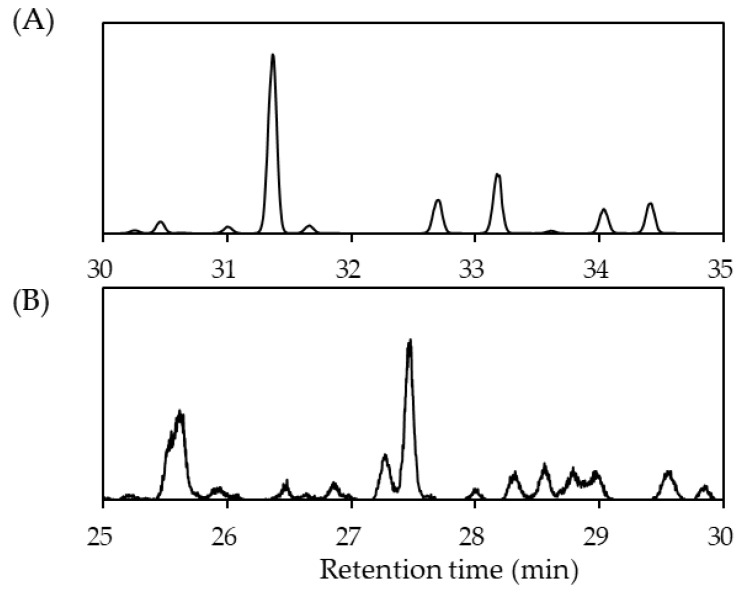
Chromatogram result for (**A**) H0-DBT; and (**B**) H18-DBT.

**Figure 3 materials-14-07613-f003:**
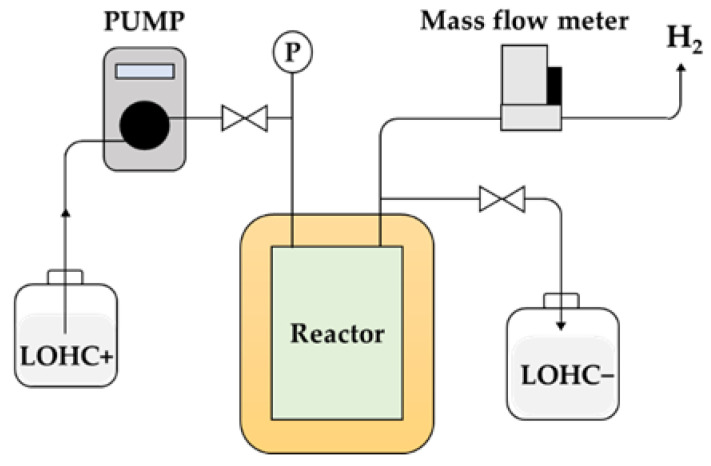
Schematic diagram of the dehydrogenation catalyst test bed (**left**) and two identical PFR reactor apparatus setup for investigating the dehydrogenation kinetics of H18-DBT (**right**).

**Figure 4 materials-14-07613-f004:**
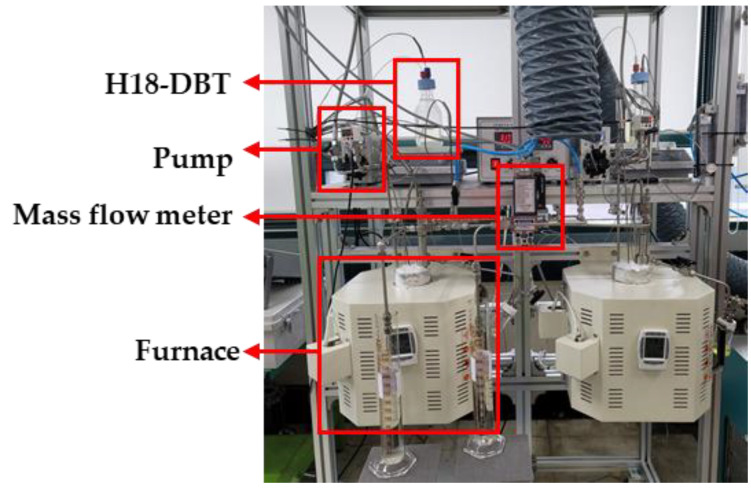
Two identical PFR reactor apparatus setups for investigating the dehydrogenation kinetics of H18-DBT.

**Figure 5 materials-14-07613-f005:**
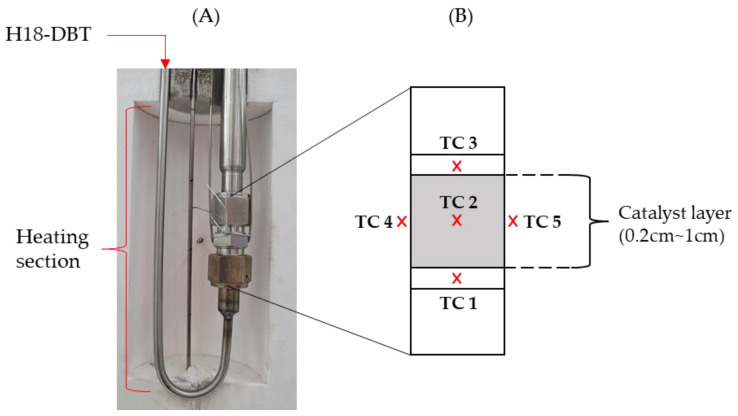
(**A**) The reactor section located within the electric furnace constituting the heating section and the catalyst bed; and (**B**) catalyst bed schematic with the marked thermocouple locations.

**Figure 6 materials-14-07613-f006:**
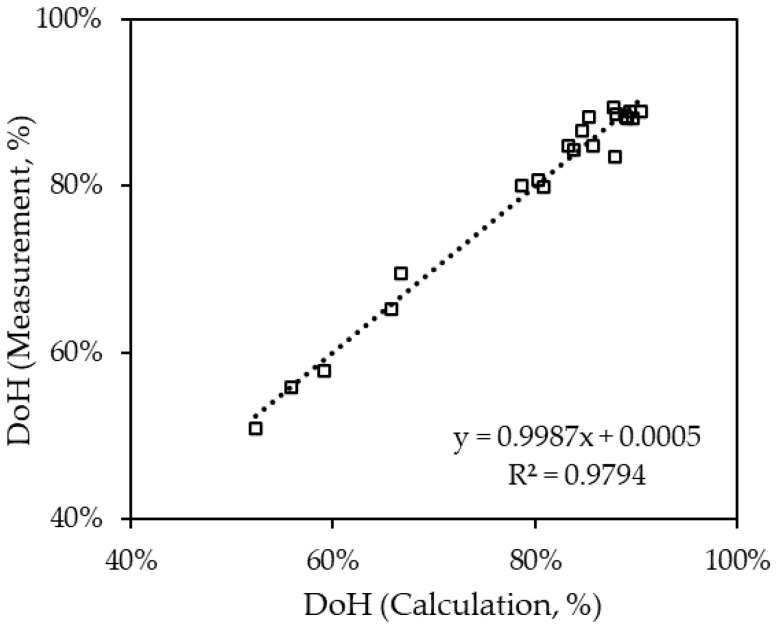
Comparison between the calculated DoH and measured DoH.

**Figure 7 materials-14-07613-f007:**
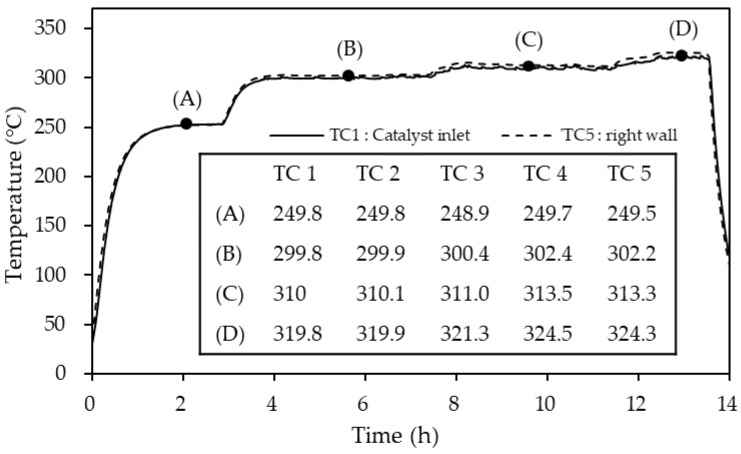
Temperature profile and values of the reactor thermocouples TC 1–TC 5 at different control temperatures of: (**A**) 250 °C; (**B**) 300 °C; (**C**) 310 °C; and (**D**) 320 °C.

**Figure 8 materials-14-07613-f008:**
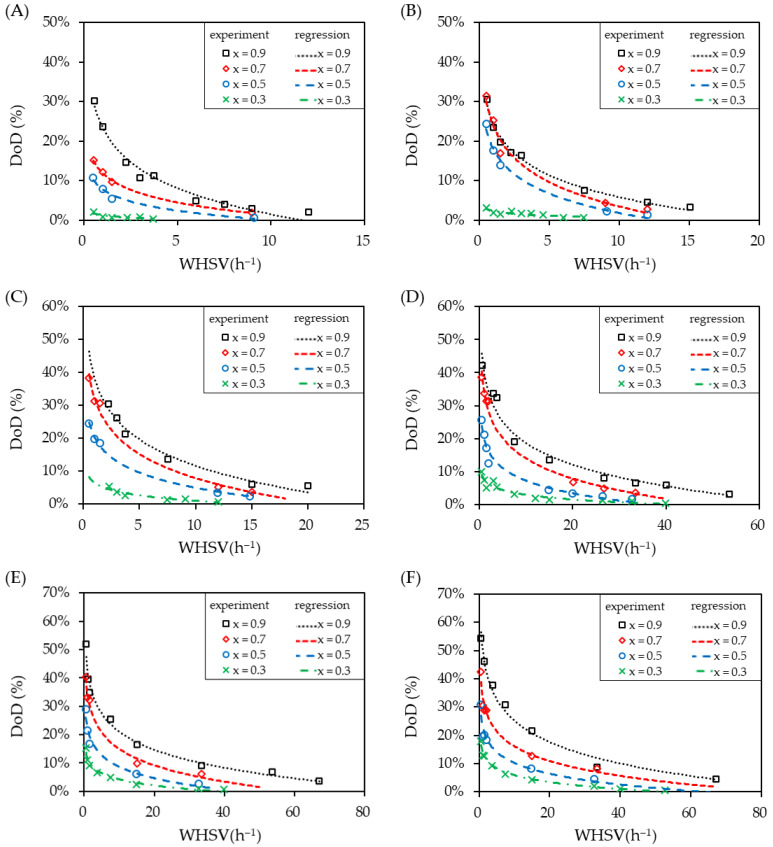
Effect of LOHC+ on the degree of dehydrogenation (DoD) due to the variations in WHSV at the initial concentration (x) and at reactor temperatures of (**A**) 270 °C; (**B**) 280 °C; (**C**) 290 °C; (**D**) 300 °C; (**E**) 310 °C; and (**F**) 320 °C.

**Figure 9 materials-14-07613-f009:**
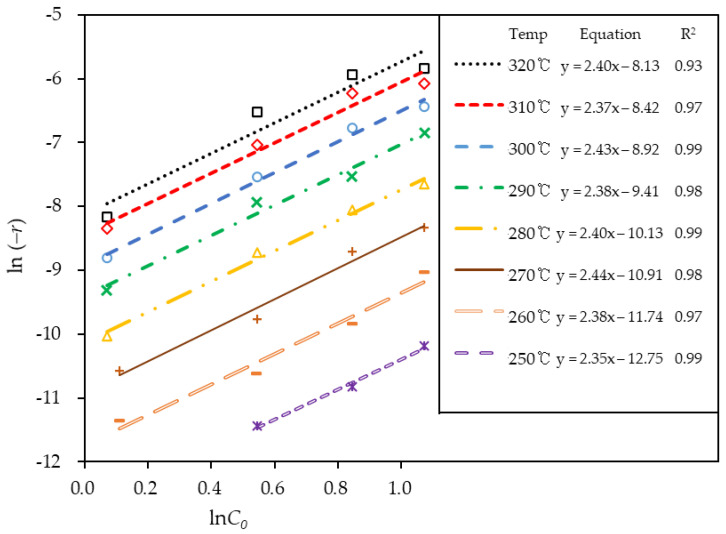
Linearized equation plot of the reaction rate (−r) and molar concentration $\left(C_{0}\right)\;$ used for evaluating the reaction order (n) and reaction constant (*K*) for the kinetic model of dehydrogenation of H18-DBT.

**Figure 10 materials-14-07613-f010:**
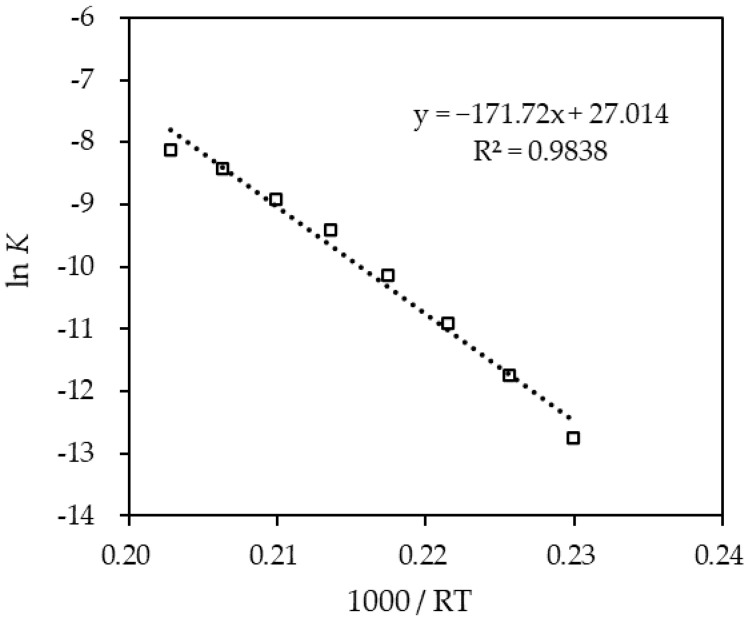
Rate constant (*K*) of the regressed experimental data following the Arrhenius approach over a dehydrogenation temperature range of 250~320 °C for H18-DBT.

**Figure 11 materials-14-07613-f011:**
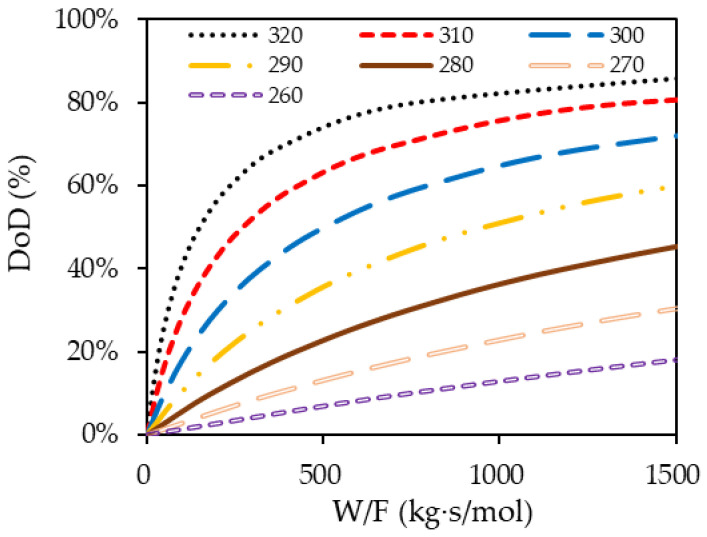
Solution of Equation (6) giving an achievable DoD for different W/F ratio at different temperatures supplied with fully hydrogenated LOHC+ in a PFR reactor.

**Figure 12 materials-14-07613-f012:**
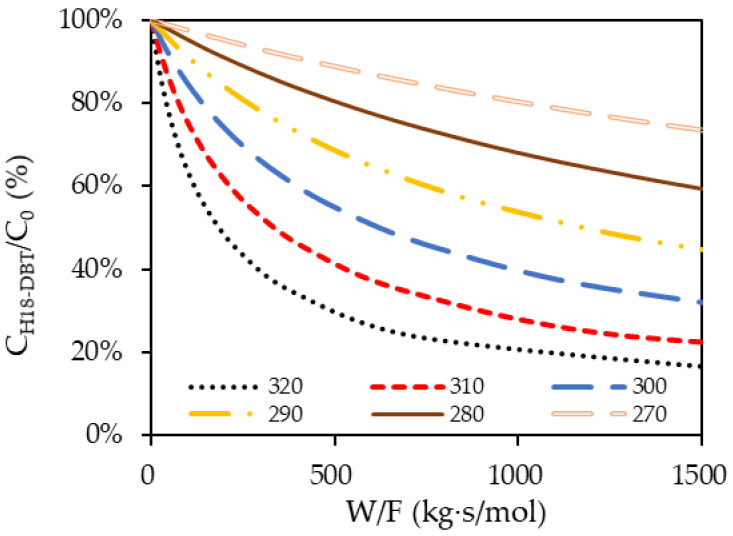
Effect of W/F on the ratio of the final molar concentration and initial molar concentration at different temperatures.

**Table 1 materials-14-07613-t001:** Description of various hydrogen production technologies [7].

Method	Raw Materials	Brief Description
Electrolysis - PV electrolysis - Photoelectrolysis	Water	Water is split into H_2_ and O_2_ by using direct current
Fossil fuel reforming - Steam reforming - Autothermal reforming	Fossil fuels	Fossil fuels are convered into CO_2_ and H_2_ by using a chemical reaction with catalyst
Thermolysis	Water	Water is thermally decomposed at high temperature
Biophotolysis	Biomass + water	Microbes or bacteria are used to generate H_2_ as a by-product
Plasma arc decomposition	Fossil fuels	Natural gas is passed through a plasma arc
Hybrid thermochemical cycles	Water	Electrical and thermal energy are used for making cyclic chemical reaction

**Table 2 materials-14-07613-t002:** Comparison of hydrogen storage technologies.

	LOHC	Liquefaction	Compression
Gravimetric hydrogen storage capacity (wt%) [13,14,15,16]	4.3~12.5	6.5–14	3–4.8
Operating temperature (°C) [13,14,15,16,17,18,19,20,21]	Ambient (storage) 25~420 (dehydrogenation)	−253	Ambient
Operating pressure (MPa) [13,22]	0.1~1	0.1	35–70
Storage time (Stability) [22]	Unlimited	Losses	Limited
Infrastructure compatibility [22]	Excellent	No	No
Reversibility [22]	Good	Not relevant	Not relevant
Process, transportation cost [22]	USD 0.5/kg H_2_ (Dibenzyl toluene)	USD 0.1~4/kg H_2_	USD 0.1~4/kg H_2_
Distribution cost [22]	-	USD 1.1~1.8/kg H_2_	USD 1.1~1.8/kg H_2_

**Table 3 materials-14-07613-t003:** Properties of LOHC materials [25].

Property	DBT	NEC	TOL
	H_0_/H_18_	H_0_/H_12_	H_0_/H_6_
Storage capacity (wt.%)	6.2	5.8	6.2
Dehydrogenation Temperature (°C)	250~320	180~270	250~450
Boiling point	407/355	270/280	111/101
Vapor pressure at 40 °C (Pa)	0.07/0.04	0.1/4.4	7880/10,900
Price (USD/kg)	4.66	46.6	0.35

(DBT: dibenzyltoluene, NEC: N-ethylcarbazole, TOL: toluene).

**Table 4 materials-14-07613-t004:** Kinetic models for the dehydrogenation of H18-DBT proposed by various researchers.

Year	Reference	Type	Catalyst (wt%)	Pressure (bar)	Temperature (°C)	Values		
						$E_{a}\;\left(kJ/mol\right)$	$k_{0}$	n
2020	Bulgarin et al. [35]	PFR	0.3% Pt–Al_2_O_3_	1	287~297	117	649,000	1
2020	Bulgarin et al. [35]	PFR	0.3% Pt–Al_2_O_3_	2.5	287~297	149	266,000,000	1
2019	Modisha et al. [36]	Batch	1% Pt–Al_2_O_3_	na	290~320	205	$e^{37.634}$	1
2019	Modisha et al. [36]	Batch	1% Pd–Al_2_O_3_	na	290~320	84	22,629	1
2019	Modisha et al. [36]	Batch	1% Pd/Pt–Al_2_O_3_	na	290~320	65	234.77	1
2019 2017	Preuster, Peters et al. [37,38]	Batch	0.5% Pt–Al_2_O_3_	1~5	260~310	119.8	125.24	1.98
2018	Wunsch et al. [39]	PFR	Pd–Ag	4	300~350	156.8 ± 28.5	$\left(2.637\pm 0.307\right)\times 10^{-6}$	variable

**Table 5 materials-14-07613-t005:** Properties of dibenzyltoluene and perhydro-dibenzyltoluene.

Property	H0-DBT	H18-DBT
Density @ 25 °C (kg/m^3^)	1.040	0.9109
Refractive index @ 25 ℃	1.602	1.493
Boiling point (°C) [26]	406.6	354.95

**Table 6 materials-14-07613-t006:** Properties of 5 wt% Pt–Al_2_O_3_ powder catalyst used in the dehydrogenation experiments.

Name	Unit	Value	Reference
BET surface area	(m^2^/g)	97, 94	[41,42]
Average pore size	(nm)	9.8, 9.2	[41,42]
Metal dispersion	(%)	22.1, 23.4	[41,42]
Metal particle diameter	(nm)	5.12, 4.8	[41,42]
Metal surface area	(m^2^/g metal)	54.5	[42]

**Table 7 materials-14-07613-t007:** Experimental conditions.

Variable	Experiment Conditions
Temperature (°C)	250, 260, 270, 280, 290, 300, 310, 320
WHSV (h^−1^)	0.5, 1, 1.5, 2, 3, 3.5, 4.5, 6, 7.5, 9, 12, 15, 20, 26, 33, 40, 52.75, 67
$x_{H18-DBT}\left(\frac{{\mathrm{mol}}_{H18-DBT}}{{\mathrm{mol}}_{\mathrm{total}}}\right)$	0.3, 0.5, 0.7, 0.9

## Data Availability

Not applicable.

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
