# Peer review of "Experimental Assessment of Perhydro-Dibenzyltoluene Dehydrogenation Reaction Kinetics in a Continuous Flow System for Stable Hydrogen Supply"

_materials, 2021, doi:10.3390/ma14247613_

Round 1

Reviewer 1 Report

Comment-1: The process that is proposed is a dehydrogenation process. The reaction that is presesnted in Figure 1 is for hydrogenation. Therefore, it is receommended to present the reaction for dehydrogenation rather than hydrogenation.

Comment-2: A comparison of different hydrogen production technolgies/methodologies against the proposed metholodogy needs to be highlighted at least based on the process conditions such as temperature, pressure and other constrainsts.

Comment-3: The purpose of this study is the dehydrogenation of perhydro-dibenzyltoluene. This is not a readily available material. The material was synthesized from dibenzyl toulne for the studies. Therefore, the cost of the production of the raw material need to considered against the raw materials from the existing methodologies.

Comment-4: There will be positional variation of concentration within a continuous packed bed reactor. The plot of ln(-r) vs. ln (Co), where Co is the feed concentration [Please refer to the rate of reaction presented in eqution no. 1 of the manuscript]. This plot is not valid because of the positional variations, it should rather be based on the exit conversion of the reactor. Also, the authors has mentioned that, the was vaporised. Clarity need to be given whether the feed concentration was taken for the vaporized stream or the liquid steam for calculations. Moreover, the rate law presented in equation (1) is a generalized power-law model which is not aplicable for catalystic dehydrogenation systems, due to adsorption, surface reaction and desoprtion steps. A different rate law and either integral approch or differntical approch needs to perfromed to find the kinetic parameters for the reaction.

Comment-5: For all the dehydogenation reaction, pressure is a critical variable in determining the degree of dehydrogenation. I recommend to study the effect of pressure on the degree of dehydrogenation of perhydro-dibenzyltoluene.

Reviewer 2 Report

In this manuscript, the authors have proposed a dehydrogenation kinetic model by studying the effect of WHSV on DoD for various initial concentrations of perhydro-dibenzyltoluene and a large range of temperature in a continuous flow system. The derived model has been applied to obtain essential information for the reactor design, which suggests the requirement of the circulation of LOHC with the catalyst several times to increase the DoD in a reactor with a limited amount of catalyst by still maintaining a low temperature.

Comment/Question: Did the authors test any other catalyst for the perhydro-dibenzyltoluene dehydrogenation reaction kinetics to see the effect of catalyst on DoD? 

Round 2

Reviewer 1 Report

In the previous review, the authors were asked to compare different technolgies of hydrogen production. Authors have not paid attention to this comment. They have compared the exsting method with different storage technologies and not the production technolgies. The article needs to be modified accordingly.

Author Response

Please see the attachemnet
